# BOUNDING MEMBERSHIP INFERENCE

## ABSTRACT

Differential Privacy (DP) is the de facto standard for reasoning about the privacy guarantees of a training algorithm. Despite the empirical observation that DP reduces the vulnerability of models to existing membership inference (MI) attacks, a theoretical underpinning as to why this is the case is largely missing in the literature. In practice, this means that models need to be trained with differential privacy guarantees that greatly decrease their accuracy. In this paper we provide a tighter bound on the accuracy of any membership inference adversary when a training algorithm provides $\epsilon$-DP. Our bound informs the design of a novel privacy amplification scheme, where an effective training set is sub-sampled from a larger set prior to the beginning of training, to greatly reduce the bound on MI accuracy. As a result, our scheme enables $\epsilon$-DP users to employ looser differential privacy guarantees when training their model to limit the success of any MI adversary; this in turn ensures that the model's accuracy is less impacted by the privacy guarantee. Finally, we discuss implications of our MI bound on machine unlearning.

## 1 INTRODUCTION

Differential Privacy (DP) (Dwork, 2006) is employed extensively to reason about privacy guarantees in a variety of different settings (Dwork, 2008). Recently, DP started being used to give privacy guarantees for the training data of deep neural networks (DNNs) learned through stochastic gradient descent (Abadi et al., 2016). However, even though DP gives privacy guarantees and bounds the worst case privacy leakage, it is not immediately clear how these guarantees bound the accuracy of known existing forms of privacy infringement attacks.

At the time of writing, the most practical attack on the privacy of DNNs is Membership Inference (MI) (Shokri et al., 2017), where an attacker predicts whether or not a model used a particular data point for training (note that this is quite similar to the hypothetical adversary at the core of the game instantiated in the definition of DP); membership inference attacks saw a strong interest by the community, and several improvements and renditions were proposed since its inception (Sablayrolles et al., 2019; Choquette-Choo et al., 2021; Maini et al., 2021; Hu et al., 2021). Having privacy guarantees would desirably defend against MI, and in fact current literature highlights that DP does indeed give an upper bound on the performance of MI adversaries (Yeom et al., 2018; Erlingsson et al., 2019; Sablayrolles et al., 2019; Jayaraman et al., 2020).

In this paper, we first propose a tighter bound on MI accuracy for training algorithms that provide $\epsilon$-DP. Our approach uses a lemma about an equivalence of certain sets of datasets. Furthermore, in obtaining our bound, we also show how this bound can benefit from a form of privacy amplification where the training dataset itself is sub-sampled from a larger dataset. Amplification is a technique pervasively found in work improving the analysis of DP learners like DP-SGD (Abadi et al., 2016), and we observe the effect of our amplification on lowering MI accuracy is significantly stronger than the effect of batch sampling, a common privacy amplification scheme for training DNNs.

Our bound also has consequences for the problem of unlearning (or data forgetting in ML) introduced by Cao & Yang (2015). In particular the MI accuracy on the point to be unlearned is a popular measure for how well a model has unlearned (Baumhauer et al., 2020; Graves et al., 2020; Golatkar et al., 2020b;a). However empirical verification of the MI accuracy can be open ended, as it is subjective to the attack employed. Theoretical bounds on all MI attacks, such as the one proposed in this work, circumvent this issue; a bound on the accuracy of MI attacks, in particular the probability a data point was used in the training dataset, indicates a limitation for any entity to discern if the

model had trained on the data point. In the case when this is sufficiently low (where sufficiently is defined apriori), one can then claim to have unlearned by achieving a model sufficiently likely to have not come from training with the data point. Our analysis shows that, if dataset subsampling is used, one can unlearn under this definition by training with a relatively large $\epsilon$ (and thus have less cost to performance).

To summarize, our contributions are:

- We present a tighter general bound on MI accuracy for $\epsilon$-DP;

- We further demonstrate how to lower this bound using a novel privacy amplification scheme built on dataset subsampling;

- We discuss the benefits of such bounds to Machine Unlearning as a rigorous way to use MI as a metric for unlearning.

## 2 BACKGROUND

### 2.1 DIFFERENTIAL PRIVACY

Differential privacy (DP) (Dwork, 2006) bounds how different the outputs of a function on adjacent inputs can be in order to give privacy guarantees for the inputs. More formally a function $F$ is $\epsilon$-DP if for all adjacent inputs $x$ and $x'$ (*i.e.* inputs with hamming distance 1) we have for all sets $S$ in the output space:

$$\mathbb{P}(F(x) \in S) \leq e^\epsilon \mathbb{P}(F(x') \in S) \tag{1}$$

We also have a more relaxed notion of $(\epsilon, \delta)$-DP where in the same setup as above, but with a parameter $\delta \in (0, 1]$, we have $\mathbb{P}(F(x) \in S) \leq e^\epsilon \mathbb{P}(F(x') \in S) + \delta$. Notably, $(\epsilon, \delta)$-DP is used for functions where it is more natural to work with $\ell_2$ metrics on the input space, which has to do with how DP guarantees are obtained.

To achieve DP guarantees one usually introduces noise to the output of the function $F$. The amount of noise is calibrated to the maximal $\ell_2$ or $\ell_1$ difference between all possible outputs of the function on adjacent datasets (also called sensitivity). Significant progress was achieved on minimizing the amount of noise needed for a given sensitivity (Balle & Wang, 2018), and on how DP guarantees scale when composing multiple DP functions (Dwork et al., 2010; Kairouz et al., 2015).

Abadi et al. (2016) demonstrated a method to make the final model returned by mini-batch SGD $(\epsilon, \delta)$-DP with respect to its training dataset by bounding the sensitivity of gradient updates during mini-batch SGD and introducing Gaussian noise to each update. This approach became the de-facto standard for DP guarantees in DNNs. However, the adoption is still greatly limited because of an observed trade-off between privacy guarantees and model utility. At the time of writing there is still no feasible ways to learn with low $\epsilon$ and high accuracy, and past work (Jagielski et al., 2020) have suggested that DP-analysis may be too loose and provides more privacy than is expected.

However, more recently Nasr et al. (2021) showed (using statistical tests and stronger MI adversaries) that the current state of the art approaches to achieving $(\epsilon, \delta)$-DP bounds for deep learning are tight, in contrast to Jagielski et al. (2020) results suggesting that they were loose. That is, there is not much more improvement to be gained by studying how to improve the $(\epsilon, \delta)$-DP bound from a given amount of noise or improving composition rules. Facing this, future improvements in DP training would lie in understanding the guarantees that DP bounds provide against the performance of relevant privacy attacks. This would allow us to be more informed about the guarantees required during training to defeat practical attacks and enable the use of looser guarantees if one is only interested in defending against a specific set of attacks. [1].

---

[1] It is worth noting that Nasr et al. (2021) showed that current analytic upper bounds on DP guarantees are tight, measuring them empirically with various strong privacy adversaries. Although results do suggest that bounds match, the paper did not investigate how DP guarantees limit performance of the adversary.

## 2.2 MEMBERSHIP INFERENCE

Shokri et al. (2017) introduced a MI attack against DNNs, which leveraged shadow models (models with the same architecture as the target model) trained on similar data in order to train a classifier which, given the outputs of a model on a data point, predicts if the model was trained on that data point or not. Since the introduction of this initial attack, the community has proposed several improved and variations of the original MI attack (Yeom et al., 2018; Salem et al., 2018; Sablayrolles et al., 2019; Truex et al., 2019; Jayaraman et al., 2020; Maini et al., 2021), such as an attack that only look at predicted labels of the target model (Choquette-Choo et al., 2021).

MI attacks are currently the main practical threat to the privacy of a user's data used to train DNNs. Especially in the context of DP DNN learning, it would be beneficial to know how DP bounds translate to bounds on the accuracy of MI attacks. This would provide an understanding on what sort of DP guarantees an entity requires to ensure a sufficiently low maximum accuracy of an adversary trying to discern their training data from their deployed model. The bounds would hence help in guiding decisions regarding setting these parameters and their corresponding guarantees.

In our paper we will work with the following (abstract) notion of MI in giving our bounds. In particular we define our MI adversary as a function $f$ which takes a set of models $S$ and a data point $\mathbf{x}^*$ to deterministically output either 1 or 0, corresponding to whether $\mathbf{x}^*$ was in the dataset $D$ used to obtain the models in $S$ or not respectively. Note the generality of this adversary, as we do not consider how the adversary obtains the function, *i.e.* it can have arbitrary strength. Any such adversary will then satisfy the upper and lower bounds we derive later in the paper.

With that definition of our adversary, we have the following definition of positive and negative MI accuracy (which is what we focus on in this paper); note that here $D$ is the training dataset and $\mathbb{P}(\mathbf{x}^* \in D|S)$ is the probability a data point $\mathbf{x}^*$ was in the training dataset used to obtain the models in the set $S$ (i.e. this is a probability over datasets). We explain more about where the randomness is introduced (in particular the probability involved in obtaining a training dataset) in Section 3.1.

**Definition 1** (MI accuracy). *The positive accuracy of $f(\mathbf{x}^*, S)$, the accuracy if $f$ outputs 1 which we define as $A(f(\mathbf{x}^*, S) = 1)$, is $\mathbb{P}(\mathbf{x}^* \in D|S)$ and the negative accuracy, the accuracy if $f$ outputs 0 which we define as $A(f(\mathbf{x}^*, S) = 0)$, is $\mathbb{P}(\mathbf{x}^* \notin D|S) = 1 - \mathbb{P}(\mathbf{x}^* \in D|S)$*

## 2.3 PREVIOUS BOUNDS

Before giving bounds on MI accuracy, we have to formally define the attack setting. Two of the main bounds (Yeom et al., 2018; Erlingsson et al., 2019) focused on an experimental setup first introduced by Yeom et al. (2018). To summarize the experiment, and in particular the situation the adversary is operating in, an adversary $f$ is given a datapoint $\mathbf{x}^*$ that is $50\%$ likely to have been used to train a model $S$ or not. The adversary then either predicts 1 if they think it was used, or 0 otherwise. Let $b = 1$ or $0$ indicate if the datapoint was or was not used for training respectively. We say the adversary was correct if their prediction matches $b$. We then define the adversary's advantage as improvement in accuracy over the $50\%$ baseline of random guessing, or more specifically $2(A(f) - 0.5)$ where $A(f)$ is the accuracy of $f$.

For such an adversary operating in a scenario where data is equally likely to be included or not in the training dataset, Yeom et al. (2018) showed that they could bound the advantage of the adversary by $e^\epsilon - 1$ when training with $\epsilon$-DP. In other words, they showed that they could bound the accuracy of the MI adversary by $e^\epsilon/2$. Their proof used the fact that the true positive rate (TPR) and false positive rate (FPR) of their adversary could be represented as expectations over the different data points in a dataset, and from that introduced the DP condition to obtain their MI bound, noting that MI advantage is equivalent to TPR - FPR.

Erlingsson et al. (2019) improved on the bound developed by Yeom et al. (2018) for an adversary operating under the same condition by utilizing a proposition given by Hall et al. (2013) on the relation between TPR and FPR for an $(\epsilon, \delta)$-DP function. Using these facts, Erlingsson et al. (2019) bounded the membership advantage by $1 - e^{-\epsilon} + \delta e^{-\epsilon}$, which is equivalent to bounding the accuracy of the adversary by $1 - e^{-\epsilon}/2$ when $\delta = 0$ (*i.e.* in $\epsilon$-DP). This is, to the best of our knowledge, the previous state-of-the-art bound for high $\epsilon$.

Other work have considered more general setups where the probability of sampling a datapoint in the dataset can vary, similar to what we consider in Section 3.1. For $\epsilon$-DP, Sablayrolles et al. (2019) bounded the probability of a datapoint $\mathbf{x}^*$ being used in the training set of a model (i.e., the accuracy of an attacker who predicted the datapoint was in the dataset of the model) by $\mathbb{P}_{\mathbf{x}^*}(1) + \frac{\epsilon}{4}$ where $\mathbb{P}_{\mathbf{x}^*}(1)$ is the probability of the datapoint being in the dataset. This is, to the best of our knowledge, the previous state-of-the-art bound for low $\epsilon$ (when reduced to case of Yeom et al. (2018) by setting $\mathbb{P}_{\mathbf{x}^*}(1) = 0.5$ as to compare with Erlingsson et al. (2019)).

Finally, Jayaraman et al. (2020) bounded the positive predictive value of an attacker (*i.e.* its precision) on a model trained with $(\epsilon, \delta)$-DP when the FPR is fixed. It is worth noting that although the considered setup is similar to works covered above, it assumes an unbalanced sampling procedure. Similarly, Jayaraman et al. further bounded membership advantage under the experiment described by Yeom et al. (2018) for a fixed FPR. Erlingsson et al. (2019) followed a similar argument for their bound but were also able to remove the need for an explicit knowledge of the FPR.

### 2.4 UNLEARNING

Having bounds on membership inference is particularly relevant to machine unlearning for DNNs. Machine unlearning was first introduced by Cao & Yang (2015), who described a setting where it is important for the model to be able to forget certain training data points and focused on the cases where there were efficient analytic solutions. It was then extended to DNNs by Bourtoule et al. (2019) with the definition that a model has unlearned a data point if after the unlearning, the distribution of models returned is identical to the one that would result from not training with the data point at all. This definition was also stated earlier by Ginart et al. (2019) for other classes of machine learning models.

Given that unlearning is interested in removing the impact a data point had on the model, further work employed MI accuracy on the data point to be unlearned as a metric for how well the model had forgotten it after using some proposed unlearning method (Baumhauer et al., 2020; Graves et al., 2020; Golatkar et al., 2020b;a). Yet, empirical estimates on the membership status of a datapoint are subjective to the concrete MI attacks employed – indeed it may be possible that there exists a stronger practical attack.

Analytic bounds to MI attacks, on the other hand, resolve the subjectivity issue of MI as a metric for unlearning as they bound the success of any adversary. In particular one could give the following definition of an unlearning guarantee from a formal MI positive accuracy bound:

**Definition 2** ($B$-MI Unlearning Guarantee). *An algorithm is $B$-MI unlearnt for $\mathbf{x}^*$ if $\mathbb{P}(\hat{\mathbf{x}}^* \notin D|S) \geq B$, i.e. the probability of $\mathbf{x}^*$ not being in the training dataset is greater than $B$.*

Therefore, our result bounding positive MI accuracy has direct consequences on the field of machine unlearning, which we further elaborate in Section 6

## 3 THE BOUND

### 3.1 THE SCENARIO

We now proceed to formalize the scenario under which our adversary operates. Here, our scenario is more general than the one introduced by Yeom et al. (2018) and, as we note later, a specific instance of it can be reduced to their setup. In particular we formalize how an entity samples data into the training dataset, and proceed with our analysis from there.

Our intuition here is that one can imagine the existence of some large data superset containing all the data points an entity could have in their training dataset. Yet, any one of these datapoints only has some probability of being sampled into the training dataset. For example, this larger dataset could consist of all the users that gave an entity access to their data, and the probability comes from the entity randomly sampling the data to use in their training dataset. This randomness can be a black-box such that not even the entity knows what data was used to train. In essence, this is the scenario Jayaraman et al. (2020) considers, though in their case, the larger dataset is implicit and takes the form of a distribution. We can then imagine that the adversary (or perhaps an arbitrator in an unlearning setup) knows the larger dataset and tries to infer whether a particular data point was

used in the training dataset. The particular MI attack that we analyze and bound is based on this scenario.

Specifically, let the individual training datasets $D$ be constructed by sampling from a finite countable set where all datapoints are unique and sampled independently, *i.e.* from some larger set $\{\mathbf{x}_1, \cdots, \mathbf{x}_N\}$. That is if $D = \{\mathbf{x}_1, \mathbf{x}_2, \cdots, \mathbf{x}_n\}$ then the probability of sampling $D$ is $\mathbb{P}(D) = \mathbb{P}_{\mathbf{x}_1}(1)\mathbb{P}_{\mathbf{x}_2}(1)\cdots\mathbb{P}_{\mathbf{x}_n}(1)\mathbb{P}_{\mathbf{x}_{n+1}}(0)\cdots\mathbb{P}_{\mathbf{x}_N}(0)$, where $\mathbb{P}_{\mathbf{x}_i}(1)$ is probability of drawing $\mathbf{x}_i$ into the dataset and $\mathbb{P}_{\mathbf{x}_i}(0)$ is the probability of not.

We define $\mathfrak{D}$ as the set of all datasets. Let now $\mathfrak{D}_{\mathbf{x}^*}$ be the set of all datasets that contain a particular point $\mathbf{x}^* \in \{\mathbf{x}_1, \cdots, \mathbf{x}_N\}$, that is $\mathfrak{D}_{\mathbf{x}^*} = \{D \ s.t \ \mathbf{x}^* \in D\}$. Similarly let $\mathfrak{D}_{\mathbf{x}^*}'$ be the set of all datasets that do not contain $\mathbf{x}^*$, *i.e.* $\mathfrak{D}_{\mathbf{x}^*}' = \{D' \ s.t. \ \mathbf{x}^* \notin D'\}$. Note $\mathfrak{D} = \mathfrak{D}_{\mathbf{x}^*} \cup \mathfrak{D}_{\mathbf{x}^*}'$ by the simple logic that any dataset has or does not have $\mathbf{x}^*$ in it. We then have the following lemma (see Appendix A for the proof).

**Lemma 1.** *$\mathfrak{D}_{\mathbf{x}^*}$ and $\mathfrak{D}_{\mathbf{x}^*}'$ are in bijective correspondence with $\mathbb{P}(D)\frac{\mathbb{P}_{\mathbf{x}^*}(0)}{\mathbb{P}_{\mathbf{x}^*}(1)} = \mathbb{P}(D')$ for $D \in \mathfrak{D}_{\mathbf{x}^*}$ and $D' \in \mathfrak{D}_{\mathbf{x}^*}'$ that map to each other under the bijective correspondence.*

Once some dataset $D$ is obtained, we call $H$ the training function which takes in $D$ and outputs a model $M$ as a set of weights in the form of a real vector. Recall that $H$ is $\epsilon$-DP if for all adjacent datasets $D$ and $D'$ and any set of model(s) $S$ in the output space of $H$ (*i.e.* some weights) we have: $\mathbb{P}(H(D) \in S) \leq e^\epsilon \mathbb{P}(H(D') \in S)$. It should be noted that from now on we assume that the set $S$ has a non-zero probability to be produced by $H$. This is sensible as we are not interested in membership inference attacks on sets of models that have 0 probability to come from training; note also if $\mathbb{P}(H(D) \in S) = 0$, then $\mathbb{P}(H(D') \in S) = 0$ for all adjacent $D'$ as $0 \leq \mathbb{P}(H(D') \in S) \leq e^\epsilon \mathbb{P}(H(D) \in S) = 0$, and thus the probability is 0 for all countable datasets as we can construct any dataset by removing and adding a data point (which does not change the probability if it is initially 0) countably many times.

## 3.2 MAIN RESULT

We now proceed to use Lemma 1 to bound the positive and negative accuracy of MI, as stated in Definition 1, for a training function $H$ that is $\epsilon$-DP under the data-sampling scenario defined earlier. Our approach differs from those we discussed in Section 2.3 in that we now focus on the definition of the conditional probability $\mathbb{P}(\mathbf{x}^* \in D|S)$ as a quotient; finding a bound then reduces to finding a way to simplify the quotient with the $\epsilon$-DP definition, which we achieve using Lemma 1.

What follows are the technical results, with Section 4, 5, and 6 discussing the main consequences of the bound.

**Theorem 1** (DP bounds MI positive accuracy). *If $f$ is a MI attack applied to a set of models $S$ and it predicts if $\mathbf{x}^*$ was in the datasets used to obtain them, and the training process $H$ is DP with $\epsilon$, its accuracy is upper-bounded by $\left(1 + \frac{e^{-\epsilon}\mathbb{P}_{\mathbf{x}^*}(0)}{\mathbb{P}_{\mathbf{x}^*}(1)}\right)^{-1}$ and lower bounded by $\left(1 + \frac{e^{\epsilon}\mathbb{P}_{\mathbf{x}^*}(0)}{\mathbb{P}_{\mathbf{x}^*}(1)}\right)^{-1}$, where $\mathbb{P}_{\mathbf{x}^*}(1)$ is the probability of drawing $\mathbf{x}^*$ into the dataset.*

See Appendix A for the proof.

By the definition of negative accuracy of $f$ we have the following corollary:

**Corollary 1** (DP bounds MI negative accuracy). *If $f$ is an MI attack applied to a set of models $S$ and it predicts if $\mathbf{x}^*$ is not in it, and the training process $H$ is DP with $\epsilon$, then the accuracy is upper-bounded by $\left(1 + \frac{e^{-\epsilon}\mathbb{P}_{\mathbf{x}^*}(1)}{\mathbb{P}_{\mathbf{x}^*}(0)}\right)^{-1}$ and lower-bounded by $\left(1 + \frac{e^{\epsilon}\mathbb{P}_{\mathbf{x}^*}(1)}{\mathbb{P}_{\mathbf{x}^*}(0)}\right)^{-1}$, where $\mathbb{P}_{\mathbf{x}^*}(1)$ is the probability of drawing $\mathbf{x}^*$ into the dataset.*

See Appendix A for the proof.

Note that in the case $\mathbb{P}_{\mathbf{x}^*}(1) = \mathbb{P}_{\mathbf{x}^*}(0) = 0.5$, the bounds given by Theorem 1 and Corollary 1 are identical. Therefore, as $f(\mathbf{x}^*, S)$ must output either 0 or 1, we have a more general claim that the attack accuracy (maximum of positive or negative accuracy) is always bounded by the same values given by Theorem 1.

# 4 THE EFFECT OF $\mathbb{P}_{\mathbf{x}^*}(1)$

We now focus on the effect of $\mathbb{P}_{\mathbf{x}^*}(1)$ which we discuss in two subsections below. The first explains how the privacy amplification, *i.e.* lowering of our MI positive accuracy bound, we observe from decreasing $\mathbb{P}_{\mathbf{x}^*}(1)$ is fundamentally different than the privacy amplification on MI from batch sampling. The second subsection outlines the practical consequences of this for a defender.

## 4.1 A NEW PRIVACY AMPLIFICATION FOR MI

Our bound given by Theorem 1 can be reduced by decreasing $\mathbb{P}_{\mathbf{x}^*}(1)$ or $\epsilon$. Furthermore, we have batch sampling, which is the probability for a data point to be in the batch for a given training step, reduces MI positive accuracy as it reduces $\epsilon$. So dataset sub-sampling ($\mathbb{P}_{\mathbf{x}^*}(1)$) and batch sampling both decrease our bound, and we term their effect "privacy amplification" (for MI) as they decrease privacy infringement (analogous to how "privacy amplification" for DP refers to methods that reduce privacy loss). We now ask the question, is the effect of dataset sub-sampling and batch sampling different?

Before proceeding, it is useful to get a sense of the impact $\mathbb{P}_{\mathbf{x}^*}(1)$ has on our bound. We plot $\mathbb{P}_{\mathbf{x}^*}(1)$ against our positive MI accuracy bound given by Theorem 1 in Figure 3 for different $\epsilon$ (see Appendix B). Notably, for a specific case when $\mathbb{P}_{\mathbf{x}^*}(1)$ is small, we get that the positive accuracy is bounded by 6.9% for $\epsilon = 2$ and $\mathbb{P}_{\mathbf{x}^*}(1) = 0.01$ (*i.e.* 1%).

We now turn to comparing the effect of batch sampling to the effect of $\mathbb{P}_{\mathbf{x}^*}(1)$ (note $\mathbb{P}_{\mathbf{x}^*}(0)/\mathbb{P}_{\mathbf{x}^*}(1) = (1 - \mathbb{P}_{\mathbf{x}^*}(1))/\mathbb{P}_{\mathbf{x}^*}(1)$). First it is worth noting that the two amplification methods are mostly independent, *i.e.* decreasing $\mathbb{P}_{\mathbf{x}^*}(0)/\mathbb{P}_{\mathbf{x}^*}(1)$ mostly places no restriction on the sampling rate for batch sizes (with some exception) [2]. Nevertheless we can ignore this restriction for the time being as we are interested in their independent mathematical behaviours. Including $q$ for DP batch privacy amplification, we can compare its impact to $\mathbb{P}_{\mathbf{x}^*}(1)$ by looking at the term $e^{-q\epsilon_0}\mathbb{P}_{\mathbf{x}}(0)/\mathbb{P}_{\mathbf{x}}(1)$ in the bound given by Theorem 1; the goal is to maximize this to make the upper bound as small as possible. In particular, we see that decreasing $q$ increases this term by $O(e^{-t})$ where as decreasing $\mathbb{P}_{\mathbf{x}^*}(1)$ increases this term by $O((1 - t)/t)$, which is slower than $O(e^{-t})$ up to a point, then faster (do note that we are looking at the order as the variable $t$ decreases). Figure 1 plots this relation, however note that the specific values are subject to change with differing constant. Nevertheless what does not change with the constants are the asymptotic behaviours, and in particular we see $\lim_{t \to 0} O((1 - t)/t) = \infty$ where as $\lim_{t \to 0} O(e^{-t}) = $ constant.

Thus we can conclude the effects of data sampling and batch sampling are different to our bound, and hence data sampling presents a new privacy amplification scheme for MI positive accuracy. As a last remark, note these same comparison holds more generally when comparing the effect of $\epsilon$ and $\mathbb{P}_{\mathbf{x}^*}(1)$;

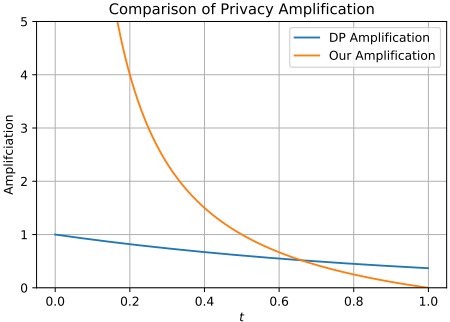

Figure 1: Comparing the DP amplification observed by decreasing batch probability (given by $e^{-t}$) to the amplification we observe from decreasing $\mathbb{P}_{\mathbf{x}}(1)$ (given by $(1 - t)/t$)

---

[2]We say "mostly" as this is true upto a point. In particular we have the expectation of the training dataset size decreases with smaller dataset sampling probabilities, and thus the lowest possible batch sampling rate $1/n$, where $n$ in the training set size, increases in expectation

### 4.2 USEFULNESS FOR A DEFENDER

We now explain one course of action a defender can take in light of this new privacy amplification for MI. In particular note that an upper bound on $\mathbb{P}_{\mathbf{x}^*}(1)$ translates to an upper bound on the relation found in Theorem 1 (as the bound is monotonically increasing with $\mathbb{P}_{\mathbf{x}^*}(1)$); hence one can, in practice, focus on giving smaller upper-bounds on $\mathbb{P}_{\mathbf{x}^*}(1)$ to decrease MI positive accuracy.

A possible approach to this is as follows. Say a user (the defender) is given some sample $D \subset \{\mathbf{x}_1, \cdots, \mathbf{x}_N\}$ drawn with some unknown distribution. In particular they do not know the individual probabilites for the points being in $D$. However say that from $D$ they obtain $D'$ by sampling any point from $D$ with probability $T$. Then the probability for any point $\mathbf{x}^* \in \{\mathbf{x}_1, \cdots, \mathbf{x}_N\}$ being in $D'$ is bounded by $T$ as the true probability is the probability $\mathbf{x}^* \in D$ (which is $\leq 1$) times $T$. Hence if the user trains with $D'$ they will have $\mathbb{P}_{\mathbf{x}^*}(1) \leq T$ and thus can use our bound to give guarantees on how low the MI positive accuracy is.

This does come with some drawbacks. In general one wants to train with more data, but by further sampling with probability $T$ we reduce our expected training dataset size. Thus a user will have to make the decision between how low they can make $T$ (in conjunction with the $\epsilon$ parameter they choose) compared to how small a dataset they are willing to train on. We leave this type of decision making for future work.

## 5 DISCUSSION

### 5.1 OUR BOUND IS TIGHTER THAN EARLIER RESULTS

We first compare our main technical result, the bound on a MI adversary's success, with the two key baselines we identified earlier (Sablayrolles et al., 2019; Erlingsson et al., 2019). The setup described in Section 3.1 is equivalent to the MI experiment defined by Yeom et al. (2018) when $\mathbb{P}_{\mathbf{x}_i}(1) = 0.5 \; \forall \mathbf{x}_i \in \{\mathbf{x}_1, \cdots, \mathbf{x}_N\}$. That is, if the training dataset was constructed by sampling data points from a larger dataset by a coin flip, then when the adversary is given any data point from the larger dataset to test there is a $50\%$ chance it was in the training set or not. Furthermore, recall that when $\mathbb{P}_{\mathbf{x}^*}(1) = 0.5$, then Theorem 1 reduces to bounds on the overall accuracy of any MI attack $f$ (as the bounds on positive and negative accuracy are the same). We can thus compare the upper-bound on MI accuracy that we achieved with the current tightest bounds given by Erlingsson et al. (2019) and Sablayrolles et al. (2019) for $\epsilon$-DP; we stated these bounds earlier in Section 2.3, where for the latter bound we also set $\mathbb{P}_{\mathbf{x}^*}(1) = 0.5$.

Our bound, and these two previous bounds, are depicted in Figure 2, where we see the bound given in Theorem 1 is always tighter than both of the previous bounds. In particular, we see that it is closer to the one introduced by Sablayrolles et al. for small $\epsilon$ and closer to the one defined by Erlingsson et al. for large $\epsilon$. Notably, for $\epsilon = 1$, we bound the accuracy of an MI attack by $73.1\%$ whereas Erlingsson et al. bound it by $81.6\%$, and Sablayrolles et al. by $75\%$. For $\epsilon = 2$, we bound MI accuracy by $88\%$ whereas Erlingsson et al. bound it by $95\%$ and Sablayrolles et al. by $100\%$.

### 5.2 ANALYSIS OF 1D LOGISTIC REGRESSION

We now motivate future work on MI bounds by illustrating how our approach of bounding the positive accuracy (defined as $\mathbb{P}(\mathbf{x}^*|S)$) of a MI adversary for $\epsilon$-DP does not immediately extend to $(\epsilon, \delta)$-DP. Such an extension would be desirable so one can capture the success of MI adversaries against training algorithms that only provide relaxed guarantees of DP, as opposed to the $\varepsilon$-DP guarantees we studied. In particular, we give a practical counter-example that shows for a specific MI attack on an $(\epsilon, \delta)$-DP logistic regression that there is no bound on the positive accuracy of the adversary, unlike what follows from our bound for $\varepsilon$-DP. Nevertheless, we demonstrate how one can still tailor bounds on general MI accuracy for this specific attack and remark how this bound is much tighter than what our theorem states for any MI adversary in the tighter $\epsilon$-DP conditions.

#### 5.2.1 POSITIVE ACCURACY IS NOT BOUNDED

Consider a set of two (scalar) points $\{x_1, x_2\}$ which are drawn into the training set $D$ with $\mathbb{P}_{x_1}(1) = 1$ and $\mathbb{P}_{x_2}(1) = 0.5$; that is $x_1$ is always in the training set, and $x_2$ has a $50\%$ chance of being in

the training set. Let model $M$ be a single dimensional logistic regression without bias defined as $M(x) = wx$ initialized such that for cross-entropy loss $L$, $\nabla L|_{\mathbf{x}_1} \approx 0$ (*i.e.* set $x_1 = \{(10^6, 1)\}$ and the weights $w = 10^6$ so that $M(x) = 10^6 x$ and thus the softmax output of the model is approximately 1 on $x_1$ and thus gradient is approximately 0). Conversely set $x_2$ such that the gradient on it is less than $-1$ (i.e., for the above setup set $x_2 = \{(10^6, 0)\}$).

Now, train the model to $(1, 10^{-5})$-DP following Abadi et al. (2016) with $\eta = 1$, sampling rate of $100\%$, a maximum gradient norm of 1, for one step. Note these are just parameters which we are allowed to change under the DP analysis, and we found the noise we would need for $(1, 10^{-5})$-DP is 4.0412. Then consider running a MI attack on the above setup where if for some threshold $\alpha$ the final weights $\mathbf{W}$ are s.t if $\mathbf{W} \le \alpha$ one classifies those weights as having come from the dataset with $x_2$, otherwise not. Do note that here we use $\mathbf{W}$ for the final weights as opposed to $w$ to emphasize that we are now talking about a random variable. The intuition for this attack is that if the dataset only contain $x_1$ then the weights do not change, but if the dataset contains $x_2$ we know the resulting gradient is negative (by construction) and thus decreases the weights (before noise).

By the earlier setup on how training datasets are constructed note that $D = \{x_1\}$ or $D = \{x_1, x_2\}$, and we will denote these $D_1$ and $D_2$ respectively. Note that if $M$ trained on $D_1$ following the suggested data points and initial weights, we have the distribution of final weight $\mathbf{W}_{D_1} = N(10^6, \sigma) = N(10^6, 4.0412)$ where $\sigma$ denotes the noise needed for $(1, 10^{-5})$-DP as stated earlier. Similarly $\mathbf{W}_{D_2} = N(10^6 - 1, 4.0412)$, since the maximum gradient norm is set to 1.

For the above MI attack we can then bound the positive accuracy as a function of $\alpha$ by:

$$\mathbb{P}(D_2 | \mathbf{W} \le \alpha) = \frac{\mathbb{P}(\mathbf{W}_{D_2} \le \alpha) * \mathbb{P}(D_2)}{\mathbb{P}(\mathbf{W}_{D_2} \le \alpha) * \mathbb{P}(D_2) + \mathbb{P}(\mathbf{W}_{D_1} \le \alpha) * \mathbb{P}(D_1)} \tag{2}$$

$$= \frac{\phi(\mathbf{W}_{D_2}, \alpha) * 0.5}{\phi(\mathbf{W}_{D_1}, \alpha) * 0.5 + \phi(\mathbf{W}_{D_2}, \alpha) * 0.5} \tag{3}$$

where $\phi(\mathbf{W}, \alpha)$ is the (Gaussian) cumulative function of random variable $\mathbf{W}$ upto $\alpha$.

We plot this in Figure 4a, and unlike Theorem 1, note how it is not bounded by anything less than 1 and goes to 1 as the threshold $\alpha$ decreases (*i.e.* $\forall m \in [0, 1)$ $\exists \alpha$ s.t $S = (-\infty, \alpha]$ yields positive accuracy greater than $m$).

### 5.2.2 MI ACCURACY IS BOUNDED

The previous section showed that for $(\epsilon, \delta)$-DP the positive accuracy of our adversary is not bounded. However, as we will show in this section, this does not mean the overall accuracy is not bounded. Specifically, for the same attack and setup as in Section 5.2.1, note that we have a bound on the general accuracy of this specific attack given by:

$$\mathbb{P}(D_1 | \mathbf{W} \ge \alpha) * \mathbb{P}(\mathbf{W} \ge \alpha) + \mathbb{P}(D_2 | \mathbf{W} \le \alpha) * \mathbb{P}(\mathbf{W} \le \alpha) = (1 - \phi(\mathbf{W}_{D_1}, \alpha)) * 0.5 + \phi(\mathbf{W}_{D_2}, \alpha) * 0.5 \tag{4}$$

We illustrate this in Figure 4b and observe that it is bounded by $54.9\%$ for $\alpha = 10^6 - 0.5$. Do note that $54.9\%$ is significantly less than $73.1\%$ which is what our bound gives for $\epsilon = 1$-DP, and $\epsilon = 1$-DP is a tighter DP condition than $(\epsilon = 1, \delta = 10^{-5})$-DP which is what is depicted in Figure 4b. This illustrates how the bound can be further tightened with a better understanding of the (worst case scenario) weight distribution and the nature of the attack. We leave this to future work.

### 5.3 MI ADVANTAGE

Previous work, particularly Yeom et al. (2018) and Erlingsson et al. (2019), focused on membership advantage, which is essentially an improvement in accuracy of $f$ over the random guess of $50\%$ of drawing a point in the training dataset. More specifically, if we let $A(f)$ denote the accuracy of $f$, then membership advantage is computed as $2(A(f) - 0.5)$. We can generalize this to ask what the membership advantage of the positive accuracy of $f$ compared to the baseline $\mathbb{P}_{\mathbf{x}^*}(1)$ is.

Theorem 1 gives us an upper bound on the positive accuracy and thus an upper bound on the positive advantage of $f$ denoted as $Ad(f)$:

$$Ad(f) = 2\left(A(f(\mathbf{x}^*), S) = 1\right) - \mathbb{P}_{\mathbf{x}^*}(1)) \le 2\left(\left(1 + \frac{e^{-\epsilon}\mathbb{P}_{\mathbf{x}^*}(0)}{\mathbb{P}_{\mathbf{x}^*}(1)}\right)^{-1} - \mathbb{P}_{\mathbf{x}^*}(1)\right) \tag{5}$$

We plotted this advantage as a function of $\mathbb{P}_{\mathbf{x}^*}(1)$ for different fixed $\epsilon$ in Figure 6 (see Appendix B). We observe that the advantage clearly depends on $\mathbb{P}_{\mathbf{x}^*}(1)$, and in fact for different $\epsilon$, the $\mathbb{P}_{\mathbf{x}^*}(1)$ resulting in the maximum advantage changes. In particular, $\mathbb{P}_{\mathbf{x}^*}(1) = 0.5$ is not close to the advantage for large $\epsilon$, which shows how the fixed experiment proposed by Yeom et al. (2018) does not necessarily give the maximum advantage an adversary could have.

However, it should be noted that higher advantage here does not mean a higher upper bound on accuracy; as we already saw in Figure 3, the upper bound on accuracy increases monotonically with $\mathbb{P}_{\mathbf{x}^*}(1)$, in contrast to the bump observed with membership advantage. This serves to contrast the study of advantage and the study of accuracy for future work.

## 6    IMPORTANCE TO DATA DELETION

The ability for an entity to decrease the ability for an arbitrator to attribute a data point to a model also has consequences for machine unlearning and data deletion requests as mentioned in Section 2.4.

In particular if $\mathbb{P}(\mathbf{x}^* \in D|S)$ is sufficiently low, that is the likelihood of $S$ coming from $\mathbf{x}^*$ is low, then an entity could claim that they do not need to delete the users data since their model is most likely independent of that data point as it most likely came from a model without it: *i.e.* leading to plausible deniability. Note we defined this type of unlearning in Section 2.4 as a $B$-MI unlearning guarantee. This is similar to the logic presented by Sekhari et al. (2021) where unlearning is presented probablistically in terms of $(\epsilon, \delta)$-unlearning.

We also observe an analogous result to Sekhari et al. (2021) where we can only handle a maximum number of deletion requests before no longer having sufficiently low probability. To be exact, let us say we do not need to undergo any unlearning process given a set of data deletion request $\hat{\mathbf{x}}^*$ if $\mathbb{P}(\hat{\mathbf{x}}^* \notin D|S) \geq B$ for some $B$ (*i.e.* we are working with probability of not having that set of data in our training set which we want to be high). Note that we sampled data independently, thus if $\hat{\mathbf{x}}^* = \{\mathbf{x}_1^*, \mathbf{x}_2^*, \cdots, \mathbf{x}_m^*\}$, then $\mathbb{P}(\hat{\mathbf{x}}^* \notin D|S) = \mathbb{P}(\mathbf{x}_1^* \notin D|S) \cdots \mathbb{P}(\mathbf{x}_m^* \notin D|S)$.

Now, for simplicity, assume the probability of drawing all points into the datasets are the same, so that for all $\mathbf{x}_i^*$ we have the same bound given by Corollary 1, that is $\mathbb{P}(\mathbf{x}_i^* \notin D|S) > L$ for some $L \leq 1$. Then we have $\mathbb{P}(\hat{\mathbf{x}}^* \notin D|S) \geq L^m$ and so an entity does not need to unlearn if $L^m \geq B$, *i.e.* if $m \leq \frac{\ln 1/B}{\ln 1/L} = \frac{\ln B}{\ln L}$. This gives a bound on how many deletion requests the entity can avoid in terms of the lower bound given in Corollary 1.

In particular, note that if $\{\mathbf{x}_1 \cdots \mathbf{x}_N\}$ is the larger set of data points an entity is sampling from, and $\mathbb{P}_{\mathbf{x}}(1) = c/N \; \forall \mathbf{x} \in \{\mathbf{x}_1 \cdots \mathbf{x}_N\}$, then the lower bound given by Corollary 1 is $\left(1 + \frac{e^{-\epsilon} \frac{c}{N}}{1 - \frac{c}{N}}\right)^{-1}$. Sekhari et al. (2021) showed that with typical DP the deletion requests grow linearly with the size of the training (in the above case $c$ represents the expected training set size). We thus compare a linear line w.r.t to $c$ to $\frac{\ln B}{\ln L(c)}$ (where $L$ is given in the earlier expression for the bound from Corollary 1) in Figure 7 (see Appendix B) to observe their respective magnitude: we fix $B = 0.8, N = 10000$ and $\epsilon = 1$ as we are interested in general trends. We observe that our deletion capacity is significantly higher for low expected training set sizes and is marginally lower than a linear trend for larger training set sizes.

## 7    CONCLUSION

In this work, we provide a tighter bound on MI accuracy against ML models trained with $\epsilon$-DP. Our bound highlights that intricacies of dataset construction are of great importance for model vulnerability to MI attacks. Indeed, based on our findings, we develop a privacy amplification scheme that just requires one to sub-sample their training dataset from larger pool of possible data points.

Based on our results, entities training their ML models with DP can employ looser privacy guarantees (and thereby preserve their models' accuracy better) while still limiting the success of MI attacks. Finally, our bound, and more generally bounds on positive MI accuracy, can also be applied to handle unlearning requests when doing machine unlearning if unlearning is defined by achieving a model with low probability of having come that data point.

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

## A  PROOFS

**Lemma 1** Note that for a given $D = \{\mathbf{x}_1, \cdots, \mathbf{x}_n\} \in \mathfrak{D}_{\mathbf{x}^*}$, $D' = D/\mathbf{x}^* \in \mathfrak{D}_{\mathbf{x}^*}'$ is unique (*i.e.* the map by removing $\mathbf{x}^*$ is injective) and similarly for a given $D' \in \mathfrak{D}_{\mathbf{x}^*}'$ $D = D' \cup \mathbf{x}^*$ is unique (*i.e.* the map by adding $\mathbf{x}^*$ is injective). Thus, we have injective maps running both ways which are the inverses of each other. As a consequence, we have $\mathfrak{D}_{\mathbf{x}^*}$ and $\mathfrak{D}_{\mathbf{x}^*}'$ are in bijective correspondence.

Now if the larger set of datapoints is $\{\mathbf{x}_1 \cdots \mathbf{x}_{n-1}, \mathbf{x}^*, \mathbf{x}_n \cdots \mathbf{x}_N\}$ letting $D = \{\mathbf{x}_1 \cdots \mathbf{x}_{n-1}\} \cup \mathbf{x}^*$ and $D' = \{\mathbf{x}_1 \cdots \mathbf{x}_{n-1}\}$ be any pair of datasets that map to each other by the above bijective map, then note $\mathbb{P}(D) = \mathbb{P}_{\mathbf{x}_1}(1)\mathbb{P}_{\mathbf{x}_2}(1)\cdots\mathbb{P}_{\mathbf{x}_{n-1}}(1)\mathbb{P}_{\mathbf{x}^*}(1)\cdots\mathbb{P}_{\mathbf{x}_{n+1}}(0)\cdots\mathbb{P}_{\mathbf{x}_N}(0)$ and $\mathbb{P}(D') = \mathbb{P}_{\mathbf{x}_1}(1)\mathbb{P}_{\mathbf{x}_2}(1)\cdots\mathbb{P}_{\mathbf{x}_{n-1}}(1)\mathbb{P}_{\mathbf{x}^*}(0)\cdots\mathbb{P}_{\mathbf{x}_{n+1}}(0)\cdots\mathbb{P}_{\mathbf{x}_N}(0)$. In particular we have $\mathbb{P}(D)\frac{\mathbb{P}_{\mathbf{x}^*}(0)}{\mathbb{P}_{\mathbf{x}^*}(1)} = \mathbb{P}(D')$.

**Theorem 1** The positive accuracy of $f$ is:

$$A(f(\mathbf{x}^*, S) = 1) = \mathbb{P}(\mathbf{x}^*|S) = \frac{\sum_{D \in \mathfrak{D}_{\mathbf{x}^*}} \mathbb{P}(H(D) \in S)\mathbb{P}(D)}{\sum_{D \in \mathfrak{D}} \mathbb{P}(H(D) \in S)\mathbb{P}(D)} \tag{6}$$

By the observation $\mathfrak{D} = \mathfrak{D}_{\mathbf{x}^*} \cup \mathfrak{D}_{\mathbf{x}^*}'$ we have that the denominator can be split into $\sum_{D \in \mathfrak{D}_{\mathbf{x}^*}} \mathbb{P}(H(D) \in S)\mathbb{P}(D) + \sum_{D' \in \mathfrak{D}_{\mathbf{x}^*}'} \mathbb{P}(H(D') \in S)\mathbb{P}(D')$.

By Lemma 1, we can replace the $D' \in \mathfrak{D}_{\mathbf{x}^*}'$ in the second sum by $D \in \mathfrak{D}_{\mathbf{x}^*}$ and replace $\mathbb{P}(D')$ by $\mathbb{P}(D)\frac{\mathbb{P}_{\mathbf{x}^*}(0)}{\mathbb{P}_{\mathbf{x}^*}(1)}$. For $\mathbb{P}(H(D') \in S)$ note by $H$ being $\epsilon$-DP we have $\mathbb{P}(H(D') \in S) \geq e^{-\epsilon}\mathbb{P}(H(D) \in S)$ and so with the previous replacements we have that the denominator is greater than $(1 + \frac{e^{-\epsilon}\mathbb{P}_{\mathbf{x}^*}(0)}{\mathbb{P}_{\mathbf{x}^*}(1)}) \cdot \sum_{D \in \mathfrak{D}_{\mathbf{x}^*}} \mathbb{P}(H(D) \in S)\mathbb{P}(D)$.

Thus, the accuracy of $f$ is $\leq \frac{1}{1 + \frac{e^{-\epsilon}\mathbb{P}_{\mathbf{x}^*}(0)}{\mathbb{P}_{\mathbf{x}^*}(1)}}$ (*i.e.* the upper bound).

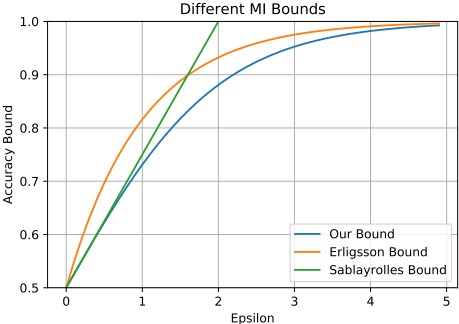

Figure 2: Comparing the upper bound to MI performance we achieved to that given by Erlingsson et al. (2019) and Sablayrolles et al. (2019) (note $\mathbb{P}_{\mathbf{x}^*}(1) = 0.5$ here). In particular note we are tighter for all $\epsilon$.

If instead we used the fact that $\mathbb{P}(H(D') \in S) \leq e^\epsilon \mathbb{P}(H(D) \in S)$, we would find that the accuracy of $f$ is $\geq \frac{1}{1 + \frac{e^\epsilon \mathbb{P}_{x^*}(0)}{\mathbb{P}_{x^*}(1)}}$ (*i.e.* the lower bound).

**Corollary 1** Immediately follows from Theorem 1 and Defintion 1, as if $A(f(\mathbf{x}^*, S) = 1) \leq \frac{1}{1 + \frac{e^\epsilon \mathbb{P}_{x^*}(0)}{\mathbb{P}_{x^*}(1)}}$ then $A(f(\mathbf{x}^*, S) = 0) = 1 - A(f(\mathbf{x}^*, S) = 1) \leq 1 - \frac{1}{1 + \frac{e^\epsilon \mathbb{P}_{x^*}(0)}{\mathbb{P}_{x^*}(1)}} = \frac{1}{1 + \frac{e^{-\epsilon} \mathbb{P}_{x^*}(1)}{\mathbb{P}_{x^*}(0)}}$.

Similarly, we get $A(f(\mathbf{x}^*, S) = 0) = 1 - A(f(\mathbf{x}^*, S) = 1) \geq 1 - \frac{1}{1 + \frac{e^{-\epsilon} \mathbb{P}_{x^*}(0)}{\mathbb{P}_{x^*}(1)}} = \frac{1}{1 + \frac{e^\epsilon \mathbb{P}_{x^*}(1)}{\mathbb{P}_{x^*}(0)}}$

## B  FIGURES

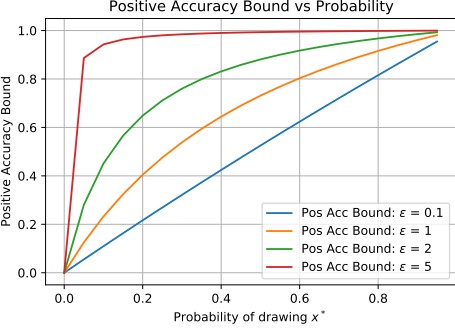

Figure 3: Our upper bound on MI positive accuracy as a function of $\mathbb{P}_{\mathbf{x}^*}(1)$

## C  TABLES

| Paper | Analytic Form | Type |
|---|---|---|
| Yeom et al. (2018) | $e^\epsilon/2$ | General |
| Erlingsson et al. (2019) | $1 - e^{-\epsilon}/2$ | General |
| Sablayrolles et al. (2019) | $\mathbb{P}_{\mathbf{x}^*}(1) + \epsilon/4$ | Positive Accuracy |
| Our Work | $\left(1 + \frac{e^{-\epsilon} \mathbb{P}_{\mathbf{x}^*}(0)}{\mathbb{P}_{\mathbf{x}^*}(1)}\right)^{-1}$ | Positive Accuracy |

Table 1: Bounds found in prior work.

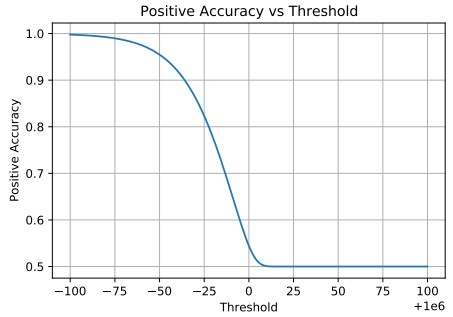 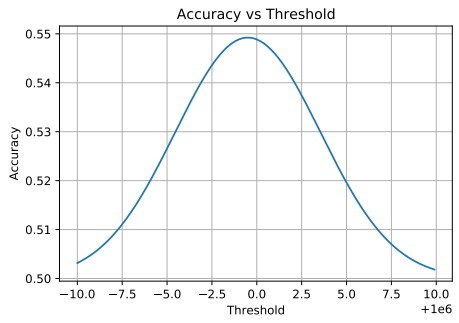

(a) Positive accuracy as a function of the threshold

(b) Accuracy as a function of the threshold

Figure 4: Impact of threshold on positive accuracy and accuracy.

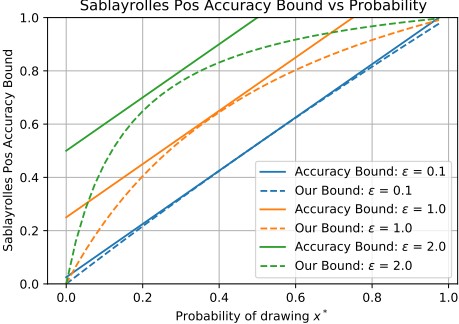

Figure 5: Sablayrolles et al. (2019) upper bound on MI positive accuracy as a function of $\mathbb{P}_{\mathbf{x}^*}(1)$ compared to our bound. Note that we are still tighter for all probabilities.

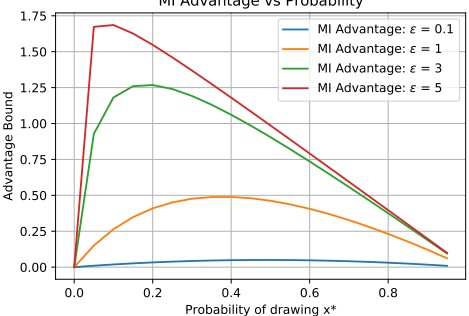

Figure 6: MI advantage

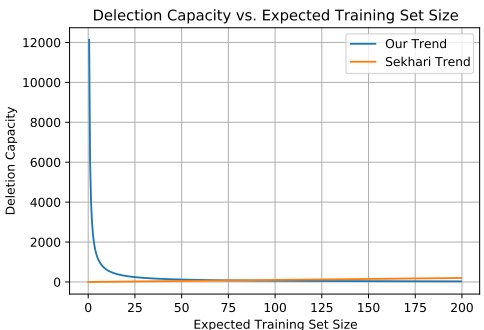

Figure 7: Comparing our deletion capacity trend to the trend Sekhari et al. (2021) describes. In particular our number of deletions degrades with training size while theirs increasing.

