# OpenReview forum: "Bounding Membership Inference"
_ICLR.cc/2022/Conference — ICLR 2022 Submitted_

### Official Review · Reviewer_gi1v · 2021-10-26

**Correctness:** 4
**Technical Novelty And Significance:** 2
**Empirical Novelty And Significance:** 2
**Recommendation:** 3
**Confidence:** 4

**Main Review:**

Edit: I have bumped my score by a point after the authors' response.

Update: on further digging into the prior work, and having discussions with other reviewers and the area chair for this paper, I have decided to downgrade my review, unfortunately. Apologies for the last minute change, but I feel like the improvements made here, which were the main focus of this paper, are a little incremental. Unless the authors’ response to the Area Chair’s comments are satisfactory, I will commit to this evaluation.

Strengths:

--The bounds for membership inference attacks are fairly novel.

--Their experiments suggest better privacy amplification as opposed to the prior work.

--The application to machine unlearning is interesting.

Weaknesses:

--This is not really a well-written paper. Sometimes there are vague blocks of texts like Lemma 1 and Definition 1. The latter, in particular, seems counter-intuitive based on what the text before that described. Also, what the probability is over isn't mentioned either. The role of $f$ in the accuracy statement isn't clear exactly.

--Nitpick, but the definition of pure DP is stated twice in the paper (Equations 1 and 2). I don't know why this could not have been done just once.

--The technical contributions don't seem that strong. The proofs don't seem to require that much depth.

--Also, their bounds on the membership inference attacks don't appear super tight. Evaluating some known attacks to compare their accuracy with their own bounds could have been some kind of an evaluation.

**Summary Of The Paper:**

This paper presents a tighter analysis for membership inference attacks on algorithms that satisfy $\varepsilon$-differential privacy. It also talks about a better form of privacy amplification based on that. Finally, it discusses the role of membership inference in the task of machine unlearning.

**Summary Of The Review:**

Based on the comments above, this paper doesn't strike me as a great paper. The ideas are respectable, but the execution could have been better, and more depth would have been appreciated.

---

> ### Author Response · Authors · 2021-11-18
> **Response to gi1v**
>
> Thank you for your feedback. In the following, we respond to your comments inline.
>
> >_** This is not really a well-written paper. Sometimes there are vague blocks of texts like Lemma 1 and Definition 1. The latter, in particular, seems counter-intuitive based on what the text before that described. Also, what the probability is over isn't mentioned either. The role of f in the accuracy statement isn't clear exactly. **_
>
> We thank the reviewer for outlining aspects of the writing that could be improved; following your guidance, we moved definition 1 into the background (Section 2.2) alongside the surrounding context. We also defined, in that same section, MI attacks and accuracy. Finally, we added more detail on what the probability is over in our definition of MI positive accuracy (which is the datasets), and also what the role of f is (which is an arbitrary function that gives a binary output given a datapoint and a set of models, and corresponds to an abstracted version of a MI attack).
>
> >_** Nitpick, but the definition of pure DP is stated twice in the paper (Equations 1 and 2). I don't know why this could not have been done just once. **_
>
> Agreed this is not ideal; we have removed the second restatement and just refer to Equation 1 in its place.
>
> >_** The technical contributions don't seem that strong. The proofs don't seem to require that much depth. **_
>
> We would like to note that our proof approach is quite different compared to prior work which relied on various statistical facts (which we described in Section 2.3). Because of this we were able to obtain the tighter bound and the novel amplification term for bounding MI. Note that now in Section 4.2 we describe how this has practical consequences for a defender, as suggested by a fellow reviewer. We elaborate on this aspect further in the following response.
>
> >_** Also, their bounds on the membership inference attacks don't appear super tight. Evaluating some known attacks to compare their accuracy with their own bounds could have been some kind of an evaluation. **_
>
> We remark that our bound is the first bound on membership inference that goes to 0 as the probability that a datapoint x is in the training dataset goes to 0 (see new Section 4.1 on asymptotic behaviour). We evaluated its significance by comparing it to prior work on bounding membership inference. In particular, Figure 5 demonstrates the advantage our bound provides over the prior work of Sablayrolles et al., which is the only directly relevant prior work that also models the probability of a datapoint x being in the training dataset. Furthermore, our bound is the first bound such that a defender can make it arbitrarily small without needing to change their DP epsilon parameter: in Section 4.2 of our revised manuscript, we now describe how a defender can thus use our bound in novel ways to control the risk of membership inference. This is done by further sampling the datasets the defender obtains (regardless of the distribution they are originally drawn from).
>
> If the reviewer has any further comments or questions, we would be happy to elaborate further.

---

> > ### Comment · Reviewer_gi1v · 2021-11-29
> > **Follow-up.**
> >
> > Thanks for your response! I have a follow-up question. How do your ideas on privacy amplification scheme differ from the previous ones? I understand that yours seem to be based on your main theorems on MI accuracy (decreasing $\mathbb{P}_{x^*}(1)$ decreases the accuracy of the MI adversary), but how are the conclusions that you draw different from previously known facts? I thought the previous results say similar things regarding the size of the subsample already.

---

> > > ### Author Response · Authors · 2021-11-29
> > > **Response to Follow-up**
> > >
> > > Thank you very much for your comments!
> > >
> > > The main difference between our and past work on privacy amplification, more precisely batch sampling privacy amplification, is that past work was focused on reducing the $\epsilon$ parameter in DP. Even though we see that reducing $\epsilon$ does reduce our bound on MI, we note that its impact is limited (see Figure 1); it cannot make the bound arbitrarily small.
> > > Dataset sampling is similar to batch sampling in the sense that it does reduce the probability for a datapoint to be included in a training batch. Hence, it can also be viewed as a form of batch sampling and so is another way of reducing $\epsilon$ for DP. However, what we observe is that in addition to reducing the $\epsilon$, it also has a completely other dynamic independent of $\epsilon$, and hence it is different from past work on DP. This new dynamic in fact shows that by reducing $\mathbb{P}_{\mathbf{x}^*}(1)$ one can make the bound on MI arbitrarily small, unlike what $\epsilon$ can do. So, though past work did also look at reducing sampling probabilities to reduce privacy leakage (i.e membership inference risk), the main difference is how much they could improve it. With our work, we now have a way of making the bound arbitrarily small. Updated section 4.1 goes through the technical details of this; please let us know if you have any further questions.

---

### Official Review · Reviewer_rm9k · 2021-11-02

**Correctness:** 4
**Technical Novelty And Significance:** 3
**Empirical Novelty And Significance:** 2
**Recommendation:** 6
**Confidence:** 3

**Main Review:**

The paper proves a tighter bound on the accuracy of any membership inference adversary when a training algorithm provides $\epsilon$-DP. Specifically, The bound is given in terms of $\mathbb{P}_{\mathbf{x}^*} (1)$, which indicates the probability of drawing
$x^*$ into the dataset from a larger set of candidates.I have verified the proofs and they are correct.

The authors further compare the amplification effects of $\mathbb{P}_{\mathbf{x}^*} (1)$ with those in DP. Besides, the author also gives the connection between the bound and data deletion in machine unlearning. Overall, the whole paper is well-written and clearly organized, and the bound provided in the paper is interesting.

My biggest question is how to use the bound to the defender in practice.  Usually, the defender does not have the prior knowledge $\mathbb{P}_{\mathbf{x}^*} (1)$, which is completely controlled by the attack. Could you please elaborate more on how to use the information of the bound when the defender who wants to to know the limit of MI accuracy when the probability of drawing
$x^*$ into the dataset is unknown to the user and how it improves from the Erlingsson et al. (2019) and Sablayrolles et al. (2019)?

Minor:

Besides, in Figure 2, is $\mathbb{P}_{\mathbf{x}^*} (1)=0.5$?

If  $\mathbb{P}_{\mathbf{x}^*} (1)\neq0.5$, does the tightness of the bound still hold? In order to do so,  I suggest the author should provide the analytical form of previous bounds of MI in the appendix.


**Summary Of The Paper:**

The paper provides a tighter bound on the accuracy of any membership inference adversary when a training algorithm provides $\epsilon$-DP. The authors also analyze their bound from the perspectives of privacy amplification schemes and its connection to machine unlearning.

**Summary Of The Review:**

1. The paper is well written and the bound is insightful.
2. The paper still needs to elaborate on the bound from the perspective of the defender.
3. More detailed comparison about the bounds of MI are needed.

---

> ### Author Response · Authors · 2021-11-15
> **Response to rm9k**
>
> Thank you for your feedback.
>
> >_** My biggest question is how to use the bound to the defender in practice. Usually, the defender does not have the prior knowledge Px∗(1), which is completely controlled by the attack. Could you please elaborate more on how to use the information of the bound when the defender who wants to to know the limit of MI accuracy when the probability of drawing x∗ into the dataset is unknown to the user and how it improves from the Erlingsson et al. (2019) and Sablayrolles et al. (2019)? **_
>
> Thank you for bringing this to our attention, we have added this specific discussion point (“usefulness for a defender”) in our now restructured section 4 (we restructured section 4 in light of suggestions given by fellow reviewers). In particular in section 4.2 we describe how all a defender needs to do to use our bound is to give an upper-bound on the probability any data point is used to train. One way the defender can obtain such a bound is by further randomly sampling the original dataset they obtained (with possibly unknown distribution).
>
> On how this relates to the previous tightest bounds, the main difference is that Erlingsson et al. does not take into account the $\mathbb{P}_{\mathbf{x}^*}(1)$ term, and though Sablayrolles does, we are always tighter than their work (see Figure 5 and our reply to your other suggestion).
>
> >_** Besides, in Figure 2, is Px∗(1)=0.5?If Px∗(1)≠0.5, does the tightness of the bound still hold? In order to do so, I suggest the author should provide the analytical form of previous bounds of MI in the appendix. **_
>
> Yes P_{x^*}(1) = 0.5 in Figure 2 (we have added this into the caption); as for the general remark on how the tightness holds when changing the probability, the main issue when trying to compare with past work is that most did not consider the case $P_{x^*}(1) \neq 0.5$. The only past work that did without fixing other constants (to the best of our knowledge) is Sablayrolles which gave the bound P_{x^*}(1) + \eps / 4 for positive accuracy. We have a plot comparing our bound with theirs for different probabilities in the appendix (see Figure 5); in particular note we are always tighter, and in fact how much tighter we are improves as $\epsilon$ increases (as for epsilon bigger than 4 their bound is always bigger than 100% which is meaningless given that we are bounding probabilities).
>
> For the comment on the analytic form of previous bounds, we had stated them in Section 2.3 when describing the past work, but we agree this is not ideal for quick reference and have thus made a table and added it to the appendix.

---

### Official Review · Reviewer_JCNw · 2021-11-03

**Correctness:** 4
**Technical Novelty And Significance:** 3
**Empirical Novelty And Significance:** Not applicable
**Recommendation:** 8
**Confidence:** 3

**Main Review:**

The main strengths of the paper are the following:
1) The authors provide a tighter bound of membership inference accuracy than existing literature. This is very consequential in privacy preserving machine learning.
2) The bounds provided by this paper are able to handle situations where the inclusion probability of samples is not 0.5 by providing separate bounds for positive and negative accuracy.
3) The authors demonstrate a benefit of their bound by connecting it to machine unlearning in a rigorous way.
4) The authors highlight the limitations of their approach by demonstrating that their bounds don't readily extend to approximate DP.

The main weaknesses of the paper are the following:
1) The background needs to substantially expanded upon. I would like to at least see the following improvements: i) a formal definition of membership inference, ii) a formal definition of machine unlearning (as used in the context of this paper).
2) I found section 4 quite difficult to follow. I believe this could be improved by presenting some background on privacy amplification by subsampling.


**Summary Of The Paper:**

The paper provides a tighter bound for accuracy of membership inference attacks for (pure) differentially private machine learning models. To account for inclusion probability of samples being different from 0.5, the authors separately bound positive and negative accuracy. Moreover, the authors also demonstrate the benefits of their bound on machine unlearning.

**Summary Of The Review:**

While the paper presents an important and novel technical contribution, it has some writing issues. In particular, the background section needs to be somewhat expanded and a particular section would benefit from more clear writing.

---

> ### Author Response · Authors · 2021-11-15
> **Response to JCNw**
>
> Thank you for your feedback.
>
> >_** The background needs to substantially expanded upon. I would like to at least see the following improvements: i) a formal definition of membership inference, ii) a formal definition of machine unlearning (as used in the context of this paper). **_
>
> We would like to thank the reviewer for these suggestions. We have now added a formal definition of Membership Inference in the context of our paper to the background; this was done by moving the previous definition 1 to the background section (with the relevant context for those statements, such as what our adversary is and what probabilities are over). We have also added a formal definition of machine unlearning in the context of this paper to the background.
>
> >_**  I found section 4 quite difficult to follow. I believe this could be improved by presenting some background on privacy amplification by subsampling. **_
>
> This was also raised by fellow reviewers, and we agree that we need to explain the context better (i.e., what privacy amplification by subsampling is, and how that then logically leads to an amplification term in our MI bound, and then what we mean by an amplification term in our MI bound context). In light of this and other suggestions, we have rewritten Section 4 to be clearer in our intentions and conclusions. This section also now contains the necessary background on privacy amplification for MI which we explain at the beginning of Section 4.1. To summarize changes made, Section 4 is now broken into two subsections: the first explains the new amplification (and what we mean by amplification and why it is new), the second explains a practical consequence of this for a defender (as suggested by a fellow reviewer).

---

### Official Review · Reviewer_rQvh · 2021-11-08

**Correctness:** 3
**Technical Novelty And Significance:** 2
**Empirical Novelty And Significance:** 2
**Recommendation:** 6
**Confidence:** 2

**Main Review:**

The presentation count be improved.  There are some parts that could be reworded to be made more clear.  Specifically, on page 1, it states “we observe the affect of our amplification on MI accuracy is significantly stronger than batch sampling”, but does this mean that batch sampling is used to amplify MI accuracy?  As far as I know, sampling is used to decrease privacy loss parameters in DP, which should reduce MI accuracy.  It is also not clear how batch sampling is different from training data that is sub-sampled.  I also do not understand the second contribution listed.  The tighter MI accuracy bound benefits in what way with subsampling?  Do you mean the bound is tighter?

Nasr et al. “Adversary Instantiation: Lower Bounds for Differentially Private Machine Learning” argues that MI is not a good attack to evaluate DP.  Do you see that increasing the privacy loss parameter with DP leads to an improved MI attack empirically, not just with theoretical bounds? As I understood it, MI was a pretty weak attack for DP, so not sure how much tightening theoretical bounds really helps here.

In the Amplification Effect section, there should be a little more care in the amplification term that is plotted.  In particular, the amplification by subsampling converts eps -> O(q eps) and the constants might matter in the plots.  This section seems a bit hand wavy in what it is trying to say.  At a high level, I understand this section to mean that subsampling amplifies DP and reducing the probability of drawing x_i in the dataset leads to an improved MI accuracy bound.  Are there two types of subsampling going on: one for subsampling the dataset and one for batch sampling.  If they are considered the same type of sampling, can you take advantage of the DP amplification, getting O(q epsilon) privacy loss, and the dataset subsample in the MI bound?

### UPDATE ###
The author feedback has addressed my concerns, so I will increase my score.

**Summary Of The Paper:**

This work studies membership inference attacks and how differential privacy can help mitigate such attacks.  Although the connection between membership inference and differential privacy has been known, this work provides a tighter connection.  It is important for the future development of private ML to better understand the connection between DP and specific attacks as small privacy loss parameters are not practical in many settings

**Summary Of The Review:**

This work provides a slightly stronger bound connecting Membership Inference attacks with differential privacy, however, the presentation could be improved and some parts are a little hand wavy.  I would also like to see more discussion on whether MIA are the right attack to consider with DP, as some recent works have shown the attack to be weak.

---

> ### Author Response · Authors · 2021-11-15
> **Response to rQvh**
>
> Thank you for your feedback.
>
> >_** The presentation count be improved. There are some parts that could be reworded to be made more clear. Specifically, on page 1, it states “we observe the effect of our amplification on MI accuracy is significantly stronger than batch sampling”, but does this mean that batch sampling is used to amplify MI accuracy? As far as I know, sampling is used to decrease privacy loss parameters in DP, which should reduce MI accuracy. It is also not clear how batch sampling is different from training data that is sub-sampled. I also do not understand the second contribution listed. The tighter MI accuracy bound benefits in what way with subsampling? Do you mean the bound is tighter? **_
>
> For the comment on page 1 we changed our phrasing to “amplification on lowering MI accuracy”; more generally, we added an explanation of what privacy amplification means for MI at the beginning of section 4.1, which is the ability to lower our MI bounds and reduce the privacy infringement. Note, that this definition is analogous to how one uses privacy amplification in the context of DP and privacy loss.
>
> Also in section 4.1 we now explain the difference between batch sampling and training data sampling in more detail. Essentially the former is about the probability of data points to be in the batch at a given step, and the latter is about the probability of data points to be in the training dataset. More importantly, we demonstrate how these two sampling procedures lead to different effects on our bounds (and hence lead to different forms of privacy amplification).
>
> Lastly, on the second contribution listed, we meant that with decreased sampling probability into the training dataset we have an even lower bound on MI accuracy. We changed the phrasing to “We further demonstrate how to lower this bound using a novel privacy amplification scheme built on dataset subsampling” which hopefully makes the statement clearer.
>
> >_** Nasr et al. “Adversary Instantiation: Lower Bounds for Differentially Private Machine Learning” argues that MI is not a good attack to evaluate DP. Do you see that increasing the privacy loss parameter with DP leads to an improved MI attack empirically, not just with theoretical bounds? As I understood it, MI was a pretty weak attack for DP, so not sure how much tightening theoretical bounds really helps here. **_
>
> On the first question about the empirical relation between DP and MI accuracy, it was shown empirically in prior work that DP can be a very strong defence against practical MI attacks (the paper “Label-Only Membership Inference Attacks” https://arxiv.org/abs/2007.14321 compared DP training to several other proposed defences, which is shown in figure 5 of that paper).
>
> On the general strength of MI as an attack against DP, it is true that the standard MI attacks are weak for DP. However, Nasr et. al. showed that one can strengthen the MI attacks significantly by using additional information available to adversaries. With this, they were able to show statistically that one can in fact instantiate an MI adversary that leads (very close) to the worst-case adversary given by DP; stated another way, MI can be a very strong adversary when they have access to more information. The catch is having access to more information, but our bound is agnostic to the information the adversary has, and thus bounds even this worse-case scenario adversary. To make clear the fact that Nasr et. al. used MI adversaries, we now explicitly state they did in section 2.1: “using statistical tests and stronger MI adversaries”.

---

> > ### Author Response · Authors · 2021-11-15
> > **Response to rQvh Continued**
> >
> > >_** In the Amplification Effect section, there should be a little more care in the amplification term that is plotted. In particular, the amplification by subsampling converts eps -> O(q eps) and the constants might matter in the plots. This section seems a bit hand wavy in what it is trying to say. At a high level, I understand this section to mean that subsampling amplifies DP and reducing the probability of drawing x_i in the dataset leads to an improved MI accuracy bound. Are there two types of subsampling going on: one for subsampling the dataset and one for batch sampling. If they are considered the same type of sampling, can you take advantage of the DP amplification, getting O(q epsilon) privacy loss, and the dataset subsample in the MI bound? **_
> >
> > In this paper we consider two different sampling procedures (sampling for a batch and sampling for the entire training dataset) and the goal of former section 4 was to outline why that is important as they lead to two different dynamics on the MI bound. It is true that constants will change the scales, however asymptotic behaviours will remain the same  -- notably our new amplification term asymptotically goes to infinity as the sampling goes to 0, whereas the batch sampling goes to some constant. That said, we believe we can emphasize the intention of that section better (with a more clear remark on these asymptotic behaviours seen in the figure), and we have re-written section 4 in light of this and other feedback given by fellow reviewers; note subsection 4.1 now concludes with how dataset sampling gives a new privacy amplification for MI.

---

### Author Response · Authors · 2021-11-19
**General Comment**

We thank the reviewers for their time, and would be happy to answer any further questions they may have before the response period ends on Monday.

---

### Comment · Area_Chair_A78o · 2021-12-04
**Updates**

This paper has generated tremendous discussion behind the scenes. I thoroughly read the entire paper myself, and we have brought in opinions from multiple other experts in the area. I wanted to summarize the current state of affairs and offer the authors the chance to understand and respond to the latest critiques. Apologies for the late nature of this comment, but as you might gather this was a long process and this is the earliest it was possible to communicate this to the authors.

The primary criticism of this paper is that the main contribution of this paper (Theorem 1) is relatively limited. In the sequel, I will try my best to discuss why and directions for improving the paper. In a following comment, I will include other minor editorial issues that I noticed in my reading (which are easy for the authors to fix, so will not affect judgment of the paper).

We can compare the magnitude of contributions in the present paper to those which contain the prior bounds (in Table 1). Yeom et al (2018) and Sablayrolles et al (2019) are conference papers, while Erlingsson et al (2019) is unpublished (though it was presented as a poster at USENIX Security). In the two conference papers, the DP -> bounds on MI accuracy results are not the focus of the papers. They seem to be tertiary results in both works (a type of brief aside, with just a few lines of simple calculations), where the primary results in these papers are much more significant. The submission's Theorem 1 is comparable to the similar results in prior works (to be discussed next), but this appears to be the largest contribution of the paper: the rest of the paper is largely about commenting on and interpreting this result.

The submission should thus be judged primarily on Theorem 1. On the positive side, as the authors point out, this is indeed a better bound than the prior works. Some other comments follow:
- The techniques used to prove Theorem 1 are fairly elementary and do not offer great novel insight into the structure of the problem. Indeed, the bijection in Lemma 1 seems to be the primary insight, which I consider to be somewhat straightforward or obvious.
- As reviewer gi1v commented, it is unclear how tight the bounds are. Is it conceivable that someone could prove a tighter bound in the exact same setting? The authors didn't really respond to this, instead commenting how the given bound improves upon prior works. In the absence of showing that the results are optimal, Theorem 1 is by definition incremental.
- Regarding the improvement on prior works: while improving on the main result of another paper is often (but not always) sufficient contribution for publication, it is generally not the case for this type of tertiary results. Therefore I do not consider the virtue of improving on the prior work in this sense to be sufficient contribution for acceptance (see also the first point above)
- The model in which the MI dataset is sampled non-uniformly was previously introduced by Sablayrolles et al (2019), so it can not be claimed as a novel contribution of this work (indeed, the authors do not, though it could be made clearer this is their model when introducing it in this paper). The authors do qualitatively improve upon their result in terms the effect of the "sampling" phenomenon.

Here are some concrete and constructive suggestions for how the paper could be strengthened.
- A "tightness" result. That is, showing that under only the guarantee of eps-DP, there exists an attack that achieves an accuracy that matches the bound in the paper.
- The paper only proves a bound for (eps, 0)-DP algorithms. But this is rather separated from the claimed motivation: in DP ML settings, (eps, delta)-DP is almost exclusively used. The authors have some discussion on these matters in Section 5.2, but it seems rather preliminary. A more thorough investigation into approximate or Renyi DP would be much more valuable. Indeed, Erlingsson et al (2019) produce some results under approximate DP, so these should be achievable in principle.
- How suited is the analysis to the specific sampling scheme? For example, what if instead of Poisson sampling, one picked a random set of a fixed size -- how would these affect the bound?
- How does generalization affect things? Empirical accuracy of an MI attack may not correspond exactly to a theoretical bound, and a good bound should take that into account.
- Why "should" we care about MI accuracy being bounded? For instance, a 51% chance of an MI attack against every member of the dataset may not be scary, but a 100% chance against 1% of the dataset would be. This very recent paper may touch on this issue (https://arxiv.org/abs/2111.09679). More broadly, how does DP affect other measures MI effectiveness?

Above are the current critiques of the paper, which we welcome the authors' perspective on, and additional directions which may strengthen the paper. Note the official metareview deadline is EoD on Monday December 6.

---

> ### Comment · Area_Chair_A78o · 2021-12-04
> **Comments and typos**
>
> Here are some notes and comments that I took during my reading of the paper. As mentioned before, these are somewhat minor and can be corrected by the authors, so will not be taken into account in the final decision of the paper.
> - The citation Dwork (2006) is not the correct one for DP: it should be Dwork, McSherry, Nissim, Smith (2006).
> - Similarly, in addition to crediting Abadi et al (2016) for DPSGD, one should also cite Song, Chaudhuri, Sarwate (2013) and Bassily, Smith, Thakurta (2014).
> - Although it is true that Jagielski et al (2020) found a gap between the success of their attacks and the guarantees provided by DP-SGD, their work doesn’t claim that this should be taken as evidence that DP analysis is too loose, so I’d suggest making it clear that this is your interpretation of their results rather than a claim in their paper.
> - Relatedly, it feels a bit disingenuous with the framing that Nasr et al (2021) refuted Jagielski et al (2020) -- the two papers really have quite a similar message, and Nasr et al (2021) really just builds on Jagielski et al (2020).
> - Membership inference has a much longer lineage than just Shokri et al (2017). Membership inference originated with Homer et al (2008)'s attack on GWAS data (http://people.eecs.berkeley.edu/~dawnsong/teaching/s10/papers/homer-resolving.pdf) and was first formalized by Sankararaman et al (2009) (https://www.nature.com/articles/ng.436).
> - Just above (1), hamming should be capitalized
> - At the bottom of page 2, there seems to be an extra . after the footnote 1.
> - The end of paragraph 1 of Section 2.2 says "look at" instead of "looks at"
> - On page 5, it may be more common to use | or : instead of s.t. in the set builder notation
> - I was a bit confused with the connection to machine unlearning -- don't most works use approximate unlearning, whereas this work focuses only on pure DP MI bounds?

---

> ### Author Response · Authors · 2021-12-05
> **Response**
>
> We appreciate the AC taking the time to reach out to us to give us a chance to comment on the current state of the internal discussion among reviewers. We also thank the AC and all of the reviewers for their time and expertise. This discussion has helped us strengthen the paper and we are thankful for that. In what follows, we will try our best to respond to all the critiques and suggestions. If any additional information is needed, we will be happy to provide further comments before the metareview deadline.
>
> >_**They seem to be tertiary results in both works (a type of brief aside, with just a few lines of simple calculations), where the primary results in these papers are much more significant. The submission's Theorem 1 is comparable to the similar results in prior works (to be discussed next), but this appears to be the largest contribution of the paper: the rest of the paper is largely about commenting on and interpreting this result.**_
>
> We would just like to briefly comment on the significance of the theorem derived in this paper, and more generally on the significance of studying bounds on MI on their own.
>
> The AC points out that all past work that had bounds on MI did so as a remark when studying some other topic: overfitting (Yeom et al), improving current attacks (Sablayrolles et al), DP (Erlingsson et al.). However, this is precisely our reason for why it now makes sense to study bounds on MI on their own; it has already been seen that such bounds help understand many different aspects of machine learning. Adding onto the previous list, we showed how bounds on MI are relevant for defenders, particularly when the bounds present ways for a defender to reduce privacy infringement, and for a formal connection between membership inference and unlearning. We have no reason to believe this list of connections is exhaustive, and in light of the scope of consequences, it is apparent that improving bounds on MI alone is quite significant. In fact, in response to a later comment on generalization we will describe how our bounds inform on the generalization gap of models. We also want to highlight that we performed a thorough analysis on how certain variables factor into the bounds, noted on the differences between various metrics of MI (positive accuracy, accuracy, advantage), and discovered the failure for certain relaxations of differential privacy to bound MI (in particular positive accuracy).
>
> To state all of this in another way, we believe the fact that many different works remark on connections with MI bounds is something that emphasises the significance of a paper devoted to studying this common theme.

---

> > ### Author Response · Authors · 2021-12-05
> > **Response Continued**
> >
> > >_**The techniques used to prove Theorem 1 are fairly elementary and do not offer great novel insight into the structure of the problem. Indeed, the bijection in Lemma 1 seems to be the primary insight, which I consider to be somewhat straightforward or obvious.**_
> >
> > On the first remark on the simplicity of the theorem, we agree that Theorem 1 is by no means a difficult proof. However, we would like to highlight the structure we gave the MI problem to make Theorem 1 follow. It may appear obvious in hindsight, yet it has several subtle (but crucial) changes to the formalisms past work on bounding MI has looked at. We think this contribution to the setup is important for future work on understanding MI.
> >
> > The first is considering some larger finite set from which training datasets are sampled. The only past work that restricted to the finite larger set, Sablayrolles et al, further assumed a specific form for the distribution of the model’s weights after training, which is stated in Equation 2 of their paper; it is clearly desirable to have bounds without such assumptions, and our bound is agnostic of them. It makes sense to consider a larger finite set as an MI adversary works with some larger dataset (the possible data points) and attempts to find which of those points were used to train a model. As such, the adversary is necessarily dealing with finite (or generally countable) sets. Though this restriction might appear like a simple change in hindsight, it is because of this reduction that we have Lemma 1 and hence our Theorem 1 (and the subsequent consequences to a defender). More generally, when dealing with distributions over some $R^n$ space, the main issue is that typically probability is defined by $fdm$, where $dm$ is lebesgue measure, and hence countable sets have measure $0$; this is to say one always has $0$ probability of having a particular datapoint in your dataset, making membership inference not particularly relevant.
> >
> > The second is the focus on positive accuracy. Once again the only work that had considered this is Sablayrolles et al., but putting aside their assumption on the distribution of the model’s weights after training, we still made significant improvements for high $\epsilon$ and low $\mathbb{P}_{\mathbf{x}^*}(1)$ (see figure 5, notably after $\epsilon =4$ their bound loses meaning as they always give above $100$\%).  The main benefit of positive accuracy, as the AC remarked when discussing “Why should we care about MI being bounded?...”, is the fact we care about the worst case scenario of knowing a datapoint was used for a model. Bounding positive accuracy bounds exactly that; by bounding positive accuracy we guarantee that an adversary only has $X$ probability of guessing a datapoint was ever used to train a model, i.e they can never have $100$\% chance of guessing a data point was used to train a model.
> >
> > Lastly, in our study of MI we made an emphasis on amplification effects as this is what is most relevant for a defender in practice. Past work mostly ignored other possible ways of reducing the bound, and in the case they did have a way (Sablayrolles et al.), their effect did not allow a defender to make the MI bound arbitrarily small which is something our bound can do (note again how our amplification is significantly better for low sampling probabilities compared to Sablayrolles et al.). It admittedly comes at a cost of having smaller training sets in expectation, but this shows that this direction is something future work can further pursue; it is also not unforeseeable that certain entities might possess far more data than they can train on, and hence benefit from our improvement on low sampling probability. We believe, beyond the proof of theorem 1 and the setup we gave that led to the proof, one of the main contributions of this paper is this amplification effect which we believe is quite novel compared to past work.

---

> > > ### Author Response · Authors · 2021-12-05
> > > **Response Continued 2**
> > >
> > > >_**As reviewer gi1v commented, it is unclear how tight the bounds are. Is it conceivable that someone could prove a tighter bound in the exact same setting? The authors didn't really respond to this, instead commenting how the given bound improves upon prior works. In the absence of showing that the results are optimal, Theorem 1 is by definition incremental.**_
> > >
> > > We do in fact know it will be tight in the limit of low or high sampling probabilities; referring to Theorem 1, we also have a lower bound on positive accuracy (which is completely absent is past work), and from this we can see that in the limit of low or high sampling probabilities they converge to the same value. Similarly, in the limit of low $\epsilon$ they converge. We are currently working on finding tightness in the middle regions; an idea we have been looking at is finding worse case probability distributions that still satisfy $\epsilon$-DP and running an MI attack on those. A notable direction for investigating tightness is the fact that though we conceived our bound in the context of ML, our theorem is not restricted to ML; hence one can abstract and look at arbitrary $\epsilon$-DP functions.
> > >
> > > >_**The model in which the MI dataset is sampled non-uniformly was previously introduced by Sablayrolles et al (2019), so it can not be claimed as a novel contribution of this work (indeed, the authors do not, though it could be made clearer this is their model when introducing it in this paper). The authors do qualitatively improve upon their result in terms the effect of the "sampling" phenomenon.**_
> > >
> > > We would just like to clarify that Sablayrolles et al. made assumptions on the distribution of the model’s weights after training which our work is agnostic to (see Equation 2 in their paper); that is, their bound is not a general bound like ours. However, putting that aside, what we changed was the linear effect of sampling given by Sablayrolles et al. to the nonlinear effect we observe, and in doing so improved quite significantly the effect of low $\mathbb{P}_{\mathbf{x}^*}(1)$ on the bound (see Figure 5).
> > >
> > > >_**The paper only proves a bound for (eps, 0)-DP algorithms. But this is rather separated from the claimed motivation: in DP ML settings, (eps, delta)-DP is almost exclusively used. The authors have some discussion on these matters in Section 5.2, but it seems rather preliminary. A more thorough investigation into approximate or Renyi DP would be much more valuable. Indeed, Erlingsson et al (2019) produce some results under approximate DP, so these should be achievable in principle.**_
> > >
> > > Note in Section 5.2.1 we showed how $\epsilon,\delta$-DP does not bound positive accuracy via a counter-example, and hence why our and future work on bounding positive accuracy needs to work with $\epsilon$-DP; note Erlingsson et al. looked at accuracy not positive accuracy and, from our earlier discussion on comments raised by the AC, positive accuracy is what we want to study. More generally, we showed the Gaussian mechanism fails to give any bounds on positive accuracy (and hence using Renyi-DP to study the privacy of Gaussian noise also fails to give bounds on positive accuracy bounds).
> > >
> > >
> > > >_**A "tightness" result. That is, showing that under only the guarantee of eps-DP, there exists an attack that achieves an accuracy that matches the bound in the paper.**_
> > >
> > > Please see our response to an earlier comment “As reviewer gi1v commented, it is unclear how tight the bounds are…” where we discuss what we know about tightness and future directions. Summarizing the discussion, we know we are tight in the limit of low and high sampling probabilities, and low $\epsilon$.
> > >
> > > >_**How suited is the analysis to the specific sampling scheme? For example, what if instead of Poisson sampling, one picked a random set of a fixed size -- how would these affect the bound?**_
> > >
> > > For that setup we have a slight issue, which is that  $D’ = D \setminus \mathbf{x}^*$ would have one less than the fixed size, so technically $\mathbb{P}(D’) = 0$  posing issues for our bijectivity lemma (in particular the mapping of probabilities part). There could be a work around, allowing one to incorporate this specific sampling into the analysis, though it would not be immediate. Questions like this highlight once again the importance of studying MI bounds on their own and understanding the effects of different sampling conditions.

---

> > > > ### Author Response · Authors · 2021-12-05
> > > > **Response Continued 3**
> > > >
> > > > >_**How does generalization affect things? Empirical accuracy of an MI attack may not correspond exactly to a theoretical bound, and a good bound should take that into account.**_
> > > >
> > > > The connection between generalization and bounding MI was investigated by Yeom et al. (see Section 3.2 in their paper where they give various bounds based on the generalization gap). To give bounds focused on generalization, Yeom et al. conceived of adversaries that would use overfitting to run MI attacks; our work is agnostic to the form of the adversary and hence does not have explicit terms relevant for specific attacks.
> > > >
> > > > However, we can in fact use their bound in conjunction with ours to determine a range on the generalization gap when training with $\epsilon$-DP; take Theorem 2 in their paper which states the MI accuracy for a specific adversary is $(R_g/B +1)/2$ (the statement of the theorem is in terms of advantage, but it is equivalent to this). We know that this must both be less than our upper bound but also greater than our lower bound when taking $P_{\mathbf{x}^*}(1) = 0.5$ (this is as our bounds are for all adversaries, and they conceived their bound in the $\mathbb{P}_{\mathbf{x}^*}(1) = 0.5$ setting), thus we have $(1 + e^{\epsilon})^{-1} \leq (R_g/B +1)/2 \leq (1 + e^{-\epsilon})^{-1}$. This is once again an application of how general work into bounding MI is, and also how it can have broad consequences for ML; the approach we showed here, which future work can further expand on, is to find bounds on specific MI adversaries dependent on the variables we are interested in, and then applying more general bounds to get a range on those variables (further tightening the general bounds then improves our knowledge on the range of the variables).
> > > >
> > > > >_**Why "should" we care about MI accuracy being bounded? For instance, a 51% chance of an MI attack against every member of the dataset may not be scary, but a 100% chance against 1% of the dataset would be. This very recent paper may touch on this issue (https://arxiv.org/abs/2111.09679). More broadly, how does DP affect other measures MI effectiveness?**_
> > > >
> > > > That is in fact precisely why we focused on positive accuracy. As mentioned in the response to the first comment, bounding positive accuracy bounds the probability of a datapoint being in the dataset used to train any model (i.e the ability of the adversary to predict any datapoint was used to train a given model), and this we believe is far more important for a defender (hence our discussion on its relevance for a defender); this also has some interesting connections to unlearning as is discussed in Section 6. This relates to the question on how DP affects other measures of MI effectiveness, i.e positive accuracy; what we know is that $\epsilon,\delta$-DP does not bound positive accuracy (we have counterexamples), whereas $\epsilon$-DP always does.
> > > >
> > > > However it should be noted, as we noted in our response to “How does generalization affect things…”, MI can be used as a general analytic tool for studying ML; in this context, past work saw a connection between generalization and MI accuracy, so more general bounds on MI accuracy allow us to understand the range of possible generalization gaps (as we showed with our bound). Further work on studying MI bounds on accuracy could lead to even tighter analysis of generalization.

---

> > > > > ### Comment · Area_Chair_A78o · 2021-12-05
> > > > > **Thanks**
> > > > >
> > > > > Thank you for the thorough response. It will be taken into account by the reviewers and myself in the final decision for your paper. Again, apologies for the last-minute nature of all of this (and on a weekend at that): I made the decision that it would be more fair to the authors to give them a chance to understand and respond to critiques and thus worth breaking the taboo of inducing work on a weekend.

---

### Decision · Program_Chairs · 2022-01-20

**Decision:**

Reject

**Comment:**

As the public post indicates, significant deliberation went into this decision. However, the core criticism remains: the primary contribution of this paper, Theorem 1, is somewhat incremental. It is acknowledged that MI is an important problem and understanding its intricacies is worthwhile, but the present paper's contributions in this space remains narrow. A more thorough exploration of the points brought up in the latest discussion and author response might help strengthen the paper. ​In particular, more careful discussion and systematic discussion and exploration of relationships between various MI attack efficacy measures (accuracy vs positive accuracy) and privacy notions (pure vs approx DP -- it wasn't totally clear how broad the result in Section 5.2.1 is without a precise theorem statement) would strengthen the paper. Additionally, while it indeed seems that the positive accuracy bound given also suffices to protect against the type of attack mentioned (where 1% of the datapoints are highly vulnerable), it is unclear if this is necessary. This feeds into the previous point: it would be valuable to get a more systematic understanding of the various MI efficacy measures and how they interact with DP. Finally, it is now appreciated that the Sablayrolles et al (2019) result worked under an unusual model restriction, though deficiencies of their result does not necessarily make this result stronger (as an aside, I believe their restriction is so that they can get a tight understanding of behavior in other settings, and DP protections were somewhat of an afterthought). The authors are encouraged to further build on this work, potentially in the directions suggested, to get a more thorough understanding of the relationships between DP and MI attacks